# Different Infectivity and Transmissibility of H5N8 and H5N1 High Pathogenicity Avian Influenza Viruses Isolated from Chickens in Japan in the 2021/2022 Season

**DOI:** 10.3390/v15020265

**Published:** 2023-01-17

**Authors:** Yoshihiro Takadate, Ryota Tsunekuni, Asuka Kumagai, Junki Mine, Yuto Kikutani, Saki Sakuma, Kohtaro Miyazawa, Yuko Uchida

**Affiliations:** 1Emerging Virus Group, Division of Zoonosis Research, National Institute of Animal Health, National Agriculture and Food Research Organization, Ibaraki 305856, Japan; 2National Veterinary Assay Laboratory, Ministry of Agriculture, Forestry and Fisheries, Tokyo 1858511, Japan

**Keywords:** high pathogenicity avian influenza virus, H5N8, H5N1, Japan, 2021/2022 season, chicken, infectivity, transmissibility

## Abstract

H5N8 and H5N1 high pathogenicity avian influenza viruses (HPAIVs) caused outbreaks in poultry farms in Japan from November 2021 to May 2022. Hemagglutinin genes of these viruses belong to clade 2.3.4.4B and can be divided phylogenetically into the following groups: 20A, 20E, and 21E. In this study, we compared the infectivity and transmissibility of HPAIVs from three groups of chickens. Representative strains from 20A, 20E, and 21E groups are A/chicken/Akita/7C/2021(H5N8)(Akita7C), A/chicken/Kagoshima/21A6T/2021(H5N1)(Kagoshima6T), and A/chicken/Iwate/21A7T/2022(H5N1)(Iwate7T), respectively. Fifty percent lethal dose of Akita7C in chickens (10^3.83^ fifty percent egg infectious dose (EID50)) was up to seven times lower than those of Kagoshima6T and Iwate7T (10^4.50^ and 10^4.68^ EID_50,_ respectively). Mean death times for Akita7C- and Kagoshima6T-infected chickens (3.45 and 3.30 days, respectively) were at least a day longer than that of Iwate7T (2.20 days). Viral titers of the trachea and cloaca of Iwate7T-infected chicken were the highest detected. The transmission rate of the Akita7C strain (100%) was markedly higher than those of the two strains (<50%). These data suggest that the infectivity and transmissibility of the Akita7C strain (H5N8) in chickens are higher than those of H5N1 viruses, providing fundamental information needed for formulating effective prevention and control strategies for HPAI outbreaks.

## 1. Introduction

Asian H5N1 subtype high pathogenicity avian influenza viruses (HPAIVs) of the A/goose/Guangdong/1/1996 (Gs/Gd) lineage have spread globally, causing huge economic losses to the poultry industry since outbreaks were reported among domestic geese in China in 1996 [1,2]. Currently, the Gs/Gd lineage is divided into 10 clades and multiple subclades, as defined by the WHO/WOAH/FAO H5 Working Group, based on viral hemagglutinin (HA) genes [3]. The neuraminidase (NA) subtype of the ancestral Gs/Gd HPAIV is N1. However, the N1 NA gene of HPAIV clade 2.3.4.4 was replaced with N2, N3, N4, N5, N6, N8, and N9 subtypes due to multiple reassortment events [2,4]. H5Nx HPAIVs belonging to clade 2.3.4.4 have caused global outbreaks since 2014 [2].

In the 2021/2022 season, outbreaks of HPAI were reported in Europe, Africa, North America, and Asia [5,6,7]. The largest number of HPAI outbreaks was reported in Europe until 2022, with H5N1 the most dominant subtype in most countries. In the 2021/2022 season (i.e., from November 2021 to May 2022) in Japan, 25 HPAI outbreaks across 12 prefectures were reported in poultry farms (chickens, ducks, and emus), and 107 cases across 8 prefectures were confirmed in wild birds (e.g., a hooded crane [8], crows, and eagles [9]) or the environment [10]. HPAIV infection was also confirmed in wild mammals (an Ezo red fox and a raccoon dog) in Hokkaido prefecture [9,11]. The number of poultry outbreaks in the 2021/2022 season was the second largest reported in Japan, followed by 52 outbreaks in the 2020/2021 season [12,13].

Sequencing analyses of HPAI strains isolated during the 2021/2022 season revealed that HPAIVs of H5N8 and H5N1 subtypes in clade 2.3.4.4B were introduced to poultry and wild birds in Japan, causing outbreaks of the season. Phylogenetic analyses based on the HA genes of all isolates from poultry revealed that the HA genes of Japanese H5N8 strains isolated in November 2021 were closely related to those of H5N8 strains of the E2 genotype isolated in Japan and Korea in 2020/2021 [8,12,14]. In contrast, the HA genes of Japanese H5N1 HPAIVs isolated in early and late periods of the 2021/2022 season (early 2021/2022: during November 2021–January 2022 and one case in May, late 2021/2022: during February 2022–May 2022) were closely related to those of H5N8 isolates in Europe in 2020/2021 and those of H5N1 isolates in Europe in 2021/2022, respectively [8,9,15] (a manuscript describing phylogenetic analyses of full genetic sequences of HPAIVs isolated in Japan during the 2021/2022 season is under preparation). Similarly, HPAI strains isolated from wild birds were divided into three groups based on genetics. Groups of H5N8 viruses isolated in November 2021 and H5N1 viruses isolated in the early and late periods of the 2021/2022 season were designated as 20A, 20E, and 21E, respectively [16]. From these data, the genetic groups of Japanese HPAI strains that caused outbreaks and wild bird infection may have shifted from viruses in the 20A and 20E groups to H5N1 viruses in the 21E group. Notably, a difference in the number of outbreaks was detected among lineages of HPAI strains (i.e., 2, 14, and 9 cases were caused by HPAI strains belonging to groups 20A, 20E, and 21E, respectively).

In general, HPAIVs exhibit high infectivity and transmissibility in chickens. However, previous studies have suggested that the infectious dose (lethal dose) and transmissibility of HPAI strains, time to death, and the amount of viral shedding from chickens markedly differ among the H5 HPAIVs belonging to clade 2.3.4.4B [13,17,18,19,20] and the differences among H5 viruses may explain the differences in numbers of outbreaks observed. In this study, we conducted animal experiments to compare the infectivity, transmissibility, time to death, and viral shedding of three genetically distinct HPAIVs isolated in Japan during the 2021/2022 season in chickens.

## 2. Materials and Methods

### 2.1. Viruses

A/chicken/Akita/7C/2021(H5N8) designated as Akita7C, A/chicken/Kagoshima/21A6T/2021(H5N1) designated as Kagoshima6T, and A/chicken/Iwate/21A7T/2022(H5N1) designated as Iwate7T were isolated by inoculating a tracheal swab or a cloacal swab collected from dead chickens at the affected farms in Akita, Kagoshima, and Iwate prefectures into 9- to 11-day-old embryonated chicken eggs. Akita7C, Kagoshima6T, and Iwate7T strains are the representative strains of the 20A, 20E, and 21E groups, respectively. All three HPAI strains were sequenced as previously described [13,19]. Nucleotide sequences of the viruses isolated in this study have been deposited in the GISAID database (http://platform.gisaid.org, accessed on 16 January 2023; Akita7C: EPI_ISL_6829532, Kagoshima6T: EPI_ISL_6829533, and Iwate7T: EPI_ISL_11147405).

### 2.2. Animal Experiments

Animal experiments were approved by the institutional committee for ethics of animal experiments (approved number 21-070 (14 October 2021)) and conducted in biosafety level 3 facilities at the National Institute of Animal Health. Animal experiments for the intravenous inoculation test, intranasal inoculation test, and transmission study were conducted as previously described [13,19]. Four-week-old and seven-week-old white leghorn chickens (L-M-6 strain) that had not previously been exposed to the influenza virus were obtained from Nisseiken (Tokyo, Japan) for use in the following animal experiments. For the intravenous inoculation test, 200 μL of a 10-fold dilution of infectious allantoic fluid of Akita7C, Kagoshima6T, and Iwate7T strains were used to intravenously inoculate eight, four- (Akita7C and Kagoshima6T) or seven-week-old chickens (Iwate7T), in accordance with WOAH criteria [21]. To calculate the 50% of the chicken lethal dose (CLD_50_) for each virus, 200 μL, diluted allantoic fluid containing 10^2^, 10^3^, 10^4^, 10^5^, and 10^6^ fifty percent egg infectious doses (EID_50_) of Akita7C and Iwate7T strains and 10^4^, 10^5^, and 10^6^ EID_50_ of Kagoshima6T strain were intranasally inoculated into five, four-week-old chickens. To assess transmissibility, one four-week-old chicken was inoculated with 10^6^ EID_50_ of Akita7C, Kagoshima6T, and Iwate7T strains. At 18 h post-inoculation (hpi), six four-week-old chickens were cohoused with the inoculated chicken. All chickens in survival and transmission studies were clinically monitored for 14 days following viral inoculation. Tracheal and cloacal swabs were collected at 1, 2, 3, 5, 7, 10, and 14 days post-inoculation (dpi) and at 0, 1, 2, 4, 6, 9, and 13 days post-cohabitation (dpc) or at death (Appendix A). Collected swabs were suspended in 2 mL Minimum Essential Medium (Thermo Fisher Scientific, Waltham, MA, United States) containing 0.5% bovine serum albumin, 25 mg/mL amphotericin B, 1000 units/mL penicillin, 1000 mg/mL streptomycin, 0.01 M HEPES, and 8.8 mg/mL NaHCO_3_ and stored in −80 ℃. Samples were used to inoculate embryonated eggs, and viral titers of each sample were determined as EID_50_ using the Reed and Muench method. Serum samples were collected from chickens before virus inoculation and from surviving chickens at 14 dpi. The Influenza A Virus Antibody Test Kit (IDEXX Laboratories, Westbrook, CT, United States) was used to detect anti-influenza virus antibodies in chicken sera in accordance with the manufacturer’s instructions.

### 2.3. Hemagglutination Inhibition (HI) Tests

Antigenic characteristics of HPAI strains were analyzed using the HI test described in the WHO manual on animal influenza diagnosis and surveillance using antiserum against HPAIVs [22]. Antisera against A/duck/Chiba/C1T/2021(H5N8) (ChibaC1T) [12], Akita7C, Kagoshima6T, and Iwate7T, were produced by immunizing chickens four times with 1.0 mg of the respective inactivated antigens (approved number 22-030 (7 July 2022)).

### 2.4. Statistical Analyses

The difference in mean death time (MDT) and viral titer in collected samples were analyzed using the Kruskal–Wallis and Steel–Dwass tests. Survival curves were compared by log-rank test corrected by the Holm method. The Kruskal–Wallis test with the Steel–Dwass test was performed by EZR software (Saitama Medical Center, Jichi Medical University, Saitama, Japan). The log-rank test adjusted by the Holm method was performed using SYSTAT version 13.2 (HULINKS, Tokyo, Japan).

## 3. Results

### 3.1. Assessment of Pathogenicity of Three Strains in Chickens

To confirm the high pathogenicity of Akita7C, Kagohima6T, and Iwate7T strains in chickens, an intravenous inoculation test was conducted according to the WOAH criteria (see Section 2.2). Because all chickens inoculated with individual strains died within 48 h (data not shown), above the 75% defined as high pathogenicity by the WOAH, they were all classified as HPAIVs. In all strains, genetic sequencing confirmed the existence of multiple basic amino acid residues in the HA cleavage site (Akita7C: PLREKRRKR/GLFG, Kagoshima6T and Iwate7T: PLRERRRKR/GLFG), a major virulence determinant of HPAIVs in chickens [23,24]. This finding also suggested the high pathogenicity of three strains considered in chickens.

### 3.2. Infectivity of Three HPAI Strains in Chickens

To estimate CLD_50_ values, MDT, and survival rates, chickens were intranasally inoculated with several doses of each HPAI strain and observed daily for 14 days (Figure 1). All chickens inoculated with 10^6^ and 10^5^ EID_50_ of the Akita7C strain died within 4.0 and 4.3 days, respectively, and three of five chickens inoculated with 10^4^ EID_50_ died within 7.3 days, whereas chickens inoculated with 10^2^ and 10^3^ EID_50_ survived throughout the observation period (Figure 1a). All chickens inoculated with 10^6^ and 10^5^ EID_50_ of the Kagoshima6T strain died within 6.0 and 9.0 days, respectively; however, chickens inoculated with 10^4^ EID_50_ survived (Figure 1b). All chickens inoculated with 10^6^ EID_50_ of the Iwate7T strain died within 3.0 days (Figure 1c). Although three of five and one of five chickens inoculated with Iwate7T strain at the doses of 10^5^ and 10^4^ EID_50_ died within 4.3 and 2.0 days, respectively, chickens inoculated with 10^3^ and 10^2^ EID_50_ survived. Anti-influenza virus antibodies were not detected in serum samples of surviving chickens, and infectious viruses were not detected from the swabs of these chickens, either. These results indicate that those chickens were not infected with HPAI strains. CLD_50_ of three strains and MDT values of chickens inoculated with each of the three strains at the dose of 10^6^ EID_50_ were calculated and summarized in Table 1. CLD_50_ values indicate that the infectivity of Akita7C in chickens is higher than those of Kagoshima6T and Iwate7T strains. The MDT of Akita7C-inoculated chickens is significantly longer than that of Iwate7T. Further, the survivability of Akita7C-infected chickens is significantly greater than that of Iwate7T.

Tracheal and cloacal swabs were collected from the virus-inoculated chickens on the time course described in Section 2.2. Viral titers of swab samples collected from chickens inoculated with 10^6^ EID_50_ viruses are shown in Figure 2. Viral titer kinetics in the trachea varied depending on the strain. Viral titer in the trachea of Akita7C-inoculated chickens moderately increased until death (Figure 2a), whereas those of Kagoshima6T- and Iwate7T-inoculated chickens rapidly increased between 1 and 2 dpi (Figure 2b,c). The mean maximum viral titer in tracheal swabs of Iwate7T-inoculated chickens was significantly higher than that in Kagoshima6T-inoculated chickens (Figure 2, Table 1). The mean maximum viral titer in cloacal swabs of Iwate7T-inoculated chickens was significantly higher than that in Akita7C- and Kagoshima6T-inoculated chickens (Figure 2, Table 1). These results demonstrate that the Iwate7T strain replicates more efficiently in both respiratory and intestinal organs than Akita7C and Kagoshima6T strains.

### 3.3. Clinical Signs Caused by Three HPAI Strains

Clinical signs of chickens infected with three HPAI strains were also compared (Table 1). Regardless of the HPAIV subtype, chickens infected with HPAI strains showed depression beginning on 2 dpi at the earliest (Akita7C: 9 of 13 chickens, Kagoshima6T: 9 of 10 chickens, Iwate7T: three of nine chickens). Clinical signs caused by HPAIV infection of the H5N8 strain (Akita7C) and the other two H5N1 strains (Kagoshima6T and Iwate7T) differed. Cyanosis in the comb and subcutaneous hemorrhage in the legs were found only in Akita7C-infected chickens (12 of 13 chickens), and these clinical signs were confirmed at 3 dpi at the earliest. Neurological symptoms such as tremors and trembling were only confirmed in chickens infected with H5N1 strains (Kagoshima6T: 2 of 10 chickens, Iwate7T: one of nine chickens).

### 3.4. Transmissibility of Three HPAI Strains in Chickens

To evaluate the transmissibility of three HPAI strains, one chicken was intranasally inoculated with each strain at the dose of 10^6^ EID_50_. Thereafter, naïve chickens were cohoused with an inoculated chicken. Clinical signs of inoculated and cohoused chickens were monitored daily for 14 days. As shown in Figure 3a, the Akita7C-inoculated chicken died on 4 dpi, meaning 3 days post-cohabitation (dpc). All chickens cohoused with the Akita7C-inoculated chicken died during the observation period. Two of six cohoused chickens died within 7.3 dpc; the remaining four chickens died within 11.0 dpc. In contrast, the Kagoshima6T-inoculated chicken died on 3 dpi (2 dpc) (Figure 3b). Half of the cohoused chickens died within 6.0 dpc, whereas the other half survived until 13 dpc. Iwate7T-inoculated chicken died on 3 dpi (Figure 3c). Two of six chickens cohoused with the Iwate7T-inoculated chicken died on 3 dpc. Anti-influenza virus antibodies and infectious viruses were not detectable in sera and swabs collected from surviving chickens, suggesting no transmission of viruses to these chickens. These results demonstrated that the transmissibility of the Akita7C strain was the highest among the three strains tested here (Table 2). Clinical signs observed in the transmission study were similar to those in the intranasal inoculation test (Table 2). 

Virus titers in tracheal and cloacal swabs were measured in the transmission study. Virus excretion to the trachea was detected in two of the six naïve chickens cohoused with an Akita7C-inoculated chicken at 4 dpc, and it was confirmed in one more chicken at 6 dpc (Figure 4a). The other three chickens excreted infectious viruses to their tracheas at 9 dpc and later. There were two viral titer peaks observed in chickens cohoused with an Akita7C inoculated chickens (4–7 dpc and 9–11 dpc). In the case of Kagoshima6T transmission, virus excretion was detected in two of six cohoused chickens at 2 dpc, with virus excretion confirmed in one more chicken at 4 dpc (Figure 4b). Virus excretion was detected in two of the six chickens cohoused with an Iwate7T-inoculated chicken on 1 dpc (Figure 4c). Mean maximum viral titer values are shown in Table 2. Notably, the replication kinetics of the three strains were similar to those determined in intranasal inoculation studies (Table 1).

### 3.5. Antigenic Analyses of H5N8 and H5N1 Strains

One of the representative HPAI strains in the 2020/2021 season, A/duck/Chiba/C1T/2021(H5N8)(ChibaC1T), which has the same gene constellation of group 20A [12], Akita7C, Kagoshima6T, and Iwate7T were antigenically analyzed by the cross HI test (Table 3). Antisera against ChibaC1T crossly reacted with Akita7C and Iwate7T antigens, but the antibody titer against Kagoshima6T was 16-fold lower than that against ChibaC1T. Antisera against Akita7C also cross-reacted with ChibaC1T and Iwate7T antigens but not with Kagoshima6T. Antisera against Kagoshima6T reacted with the only homologous antigen. Antisera against Iwate7T reacted with the other three antigens.

## 4. Discussion

H5Nx HPAIVs of clade 2.3.4.4 have caused huge poultry industry losses [2]. The second largest HPAI outbreak in Japan occurred during the 2021/2022 season, after the initial Gs/GD H5 HPAIV outbreak in 2004. Phylogenetic analyses revealed that HPAI strains in the 2021/2022 season belong to clade 2.3.4.4B and could be divided into three distinct genetic groups (20A, 20E, and 21E) [8,9,16]. Interestingly, most of the outbreaks (23 out of the 25 cases) were caused by H5N1 HPAIVs, whereas only two cases were caused by H5N8 viruses. We postulated that HPAI strains of each group had differing infectivities and transmissibilities in chickens, with differences between groups likely affecting the number of outbreaks they caused. In this study, when three strains were compared, we found that Akita7C (H5N8) had the highest infectivity in chickens among the three strains. Importantly, the CLD_50_ of the Akita7C strain was up to seven times lower than those of H5N1 strains in chickens (Table 1). We also found that Akita7C-infected chickens survived longer than Iwate7T-infected chickens (Table 1, Figure 1) since the MDT value of chickens infected with Akita7C at the dose of 10^6^ EID_50_ was more than 1 day longer than those of the others. We also found that Akita7C had the highest transmissibility among the three strains (Table 2, Figure 3). These results indicate that viruses in the 20A group (representative strain: Akita7C) may have caused many outbreaks occurring in the 2021/2022 season. However, the actual number of outbreaks caused by the 20A group was very limited. Thus, factors other than infectivity, MDT, and transmissibility may have affected the number of outbreaks caused by group 20A viruses.

Since Japan applies a stamping-out policy when H5 or H7 influenza viruses are detected in poultry farms, the incursion of viruses from wild birds rather than the transmission of viruses among poultry farms may affect the number of outbreaks that occur. Thus, the prevalence of HPAIVs in wild bird populations may affect the number of outbreaks. Indeed, the number of H5N8 viruses detected in wild birds was smaller than that of H5N1 viruses in Japan and Korea during the 2021/2022 season [16]. The low prevalence of H5N8 strains in the wild bird population might be related to the limited number of cases caused by group 20A viruses. According to a previous study [25], humoral immunity against avian influenza virus strains with similar antigenicity modulates reinfection in mallard ducks. Our analysis revealed that antisera against ChibaC1T(H5N8), which caused outbreaks in Japan in the 2020/2021 season and contains a gene constellation matching that of 20A [8,12], cross-reacted with the Akita7C antigen but not with Kagoshima6T (Table 3). Thus, antibody responses against ChibaC1T in migratory birds may have reduced the prevalence of group 20A viruses in the 2021/2022 season. Antibodies against ChibaC1T cross-reacted with Iwate7T antigen (Table 3), although the number of outbreaks caused by 21E viruses was not limited. Outbreaks in poultry farms and the detection of wild birds infected with group 21E viruses were confirmed only in northern parts of Japan (i.e., Hokkaido, Aomori, Akita, Iwate, and Miyagi prefectures) from February to May 2022 [16]. Since it has been shown that migratory bird species (e.g., eagle, swan, wigeon, and pintail) that fly to northern parts of Japan differ from those coming to other regions (the central and southern parts of Japan) [9], those not (or incompletely) exposed H5N8 viruses likely brought H5N1 viruses of group 21E to resident birds (e.g., jungle crows) that subsequently transmitted viruses to poultry. In addition, the infectivity and transmissibility of these strains in ducks may have affected the number of outbreaks that occurred. Previous studies demonstrated that survival rates and MDT values of HPAIV-inoculated ducks and levels of virus excretion differ widely among H5 viruses [18,26]. Three strains’ infectivity and transmissibility in ducks should be determined in future studies.

We also discussed differences in the infectivity and transmissibility of the three HPAI strains. We found that Akita7C has the highest infectivity among the three strains (Table 1, Figure 1). Since low infectivity and transmissibility were confirmed in several clade 2.3.4.4B HPAI strains, of which several gene segments were derived from wild birds [19,27,28,29], it is possible that Kagoshima6T and Iwate7T are not well-adapted to chickens. We also found that the Iwate7T strain replicates efficiently in both respiratory and intestinal organs compared to Akita7C and Kagoshima6T strains (Table 1, Figure 2). Then, we compared amino acid sequences functioning as pathogenicity determinants of influenza viruses among three strains [30] and found an amino acid difference in the PB1-F2 protein (Iwate7T: S66, Akita7C, and Kagoshima6T: N66). A mutation in the protein (N66S) has been shown to increase viral titers and cytokine expression in mice infected with the 1918 pandemic virus and H5N1 HPAIV [31,32]. Further, PB1-F2 expression has been shown to enhance H5N1 HPAIV infectivity in the intestinal organs of chickens [33]. It would be interesting to investigate whether PB1-F2 S66 determines efficient replication of the Iwate7T strain in the chicken intestine. We also confirmed that Akita7C’s transmissibility is the highest among the three strains (Table 2, Figure 3). High levels of infectivity and a long duration of virus shedding commonly promote efficient transmission in chickens, as described previously [19,34]. The high infectivity and long MDT of the Akita7C strain may explain its high transmissibility in chickens. However, when assessing viral transmission, two virus-shedding peaks were confirmed in Akita7C-infected chickens, suggesting that three chickens were infected via transmission, and the other chickens were infected due to subsequent transmission from the first three chickens. Thus, the actual transmission rate of the Akita7C strain should be addressed in future studies.

In this study, we demonstrated that Akita7C has greater levels of infectivity and transmissibility in chickens than the other strains tested, although all three strains were classified as HPAIVs based on the WOAH definition. Since the characteristics of HPAI strains in the 2021/2022 season were not correlated with the number of outbreaks observed, relationships between the number of outbreaks and other factors, such as the prevalence of HPAIVs in wild birds, including residential birds [35], should be addressed in future studies. In September 2022, H5N1 HPAIV was detected in a wild peregrine falcon, the earliest wild bird case since 2004 in Japan. Genetic analyses showed that the constellation of all gene segments was the same as that in group 20E, indicating that the virus, which was prevalent in the 2021/2022 season, was reintroduced the following year (GISAID ID: EPI_ISL_15923322) [36]. Viral characteristics in chickens, such as infectivity, lethality, and transmissibility of HPAI strains; MDT and clinical signs of HPAIV-infected chickens; and viral titers shed from the HPAIVs’ infected chickens have the potential to provide fundamental information needed for planning effective strategies against future HPAI outbreaks.

## Figures and Tables

**Figure 1 viruses-15-00265-f001:**
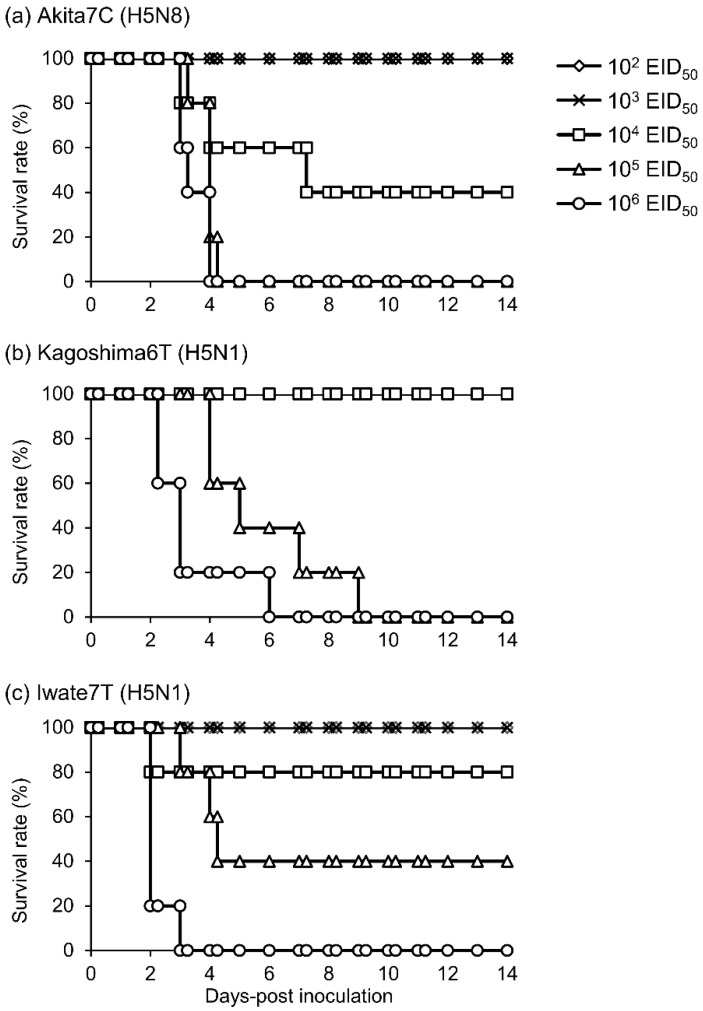
Survival rates of chickens inoculated with (**a**) Akita7C, (**b**) Kagoshima6T, and (**c**) Iwate7T strains. The survival rates of chickens inoculated with 10^2^, 10^3^,10^4^, 10^5^, and 10^6^ EID_50_ of each virus are shown by rhombuses, crosses, squares, triangles, and circles, respectively.

**Figure 2 viruses-15-00265-f002:**
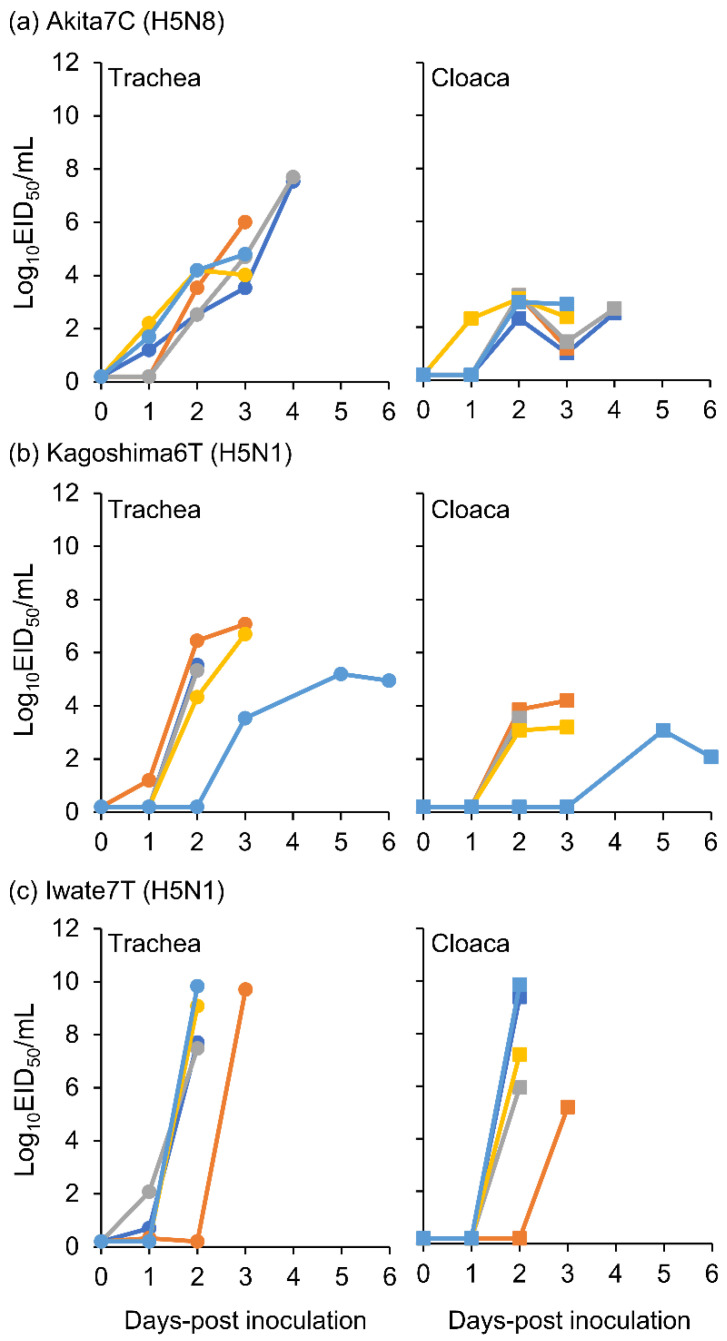
Viral titers in each tracheal and cloacal swab collected from chickens inoculated with the 10^6^ EID_50_ of (**a**) Akita7C, (**b**) Kagoshima6T, and (**c**) Iwate7T strains. Viral titers in tracheal and cloacal swabs are represented by circles and squares (blue, light blue, orange, yellow, and gray), respectively.

**Figure 3 viruses-15-00265-f003:**
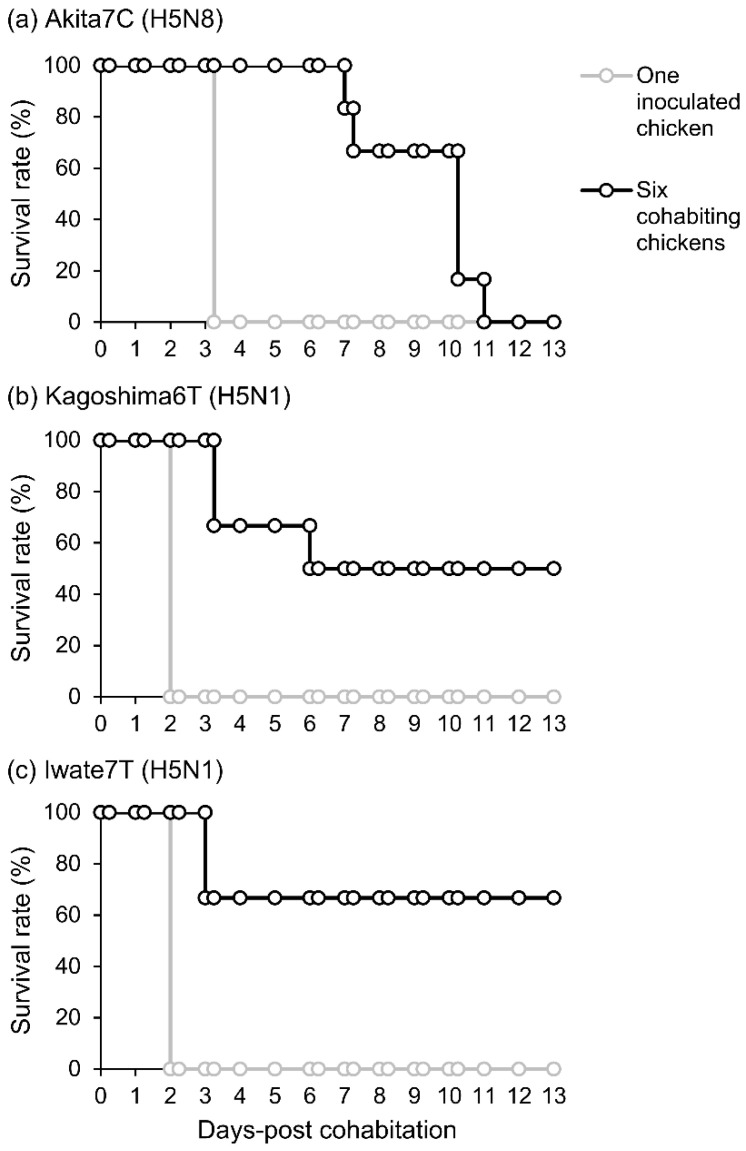
Survival rates of chickens intranasally inoculated with 10^6^ EID_50_ of (**a**) Akita7C, (**b**) Kagoshima6T, and (**c**) Iwate7T strains and cohabitated chickens. The survival rates of the virus-inoculated chicken and cohoused chickens are shown by black and white circles, respectively.

**Figure 4 viruses-15-00265-f004:**
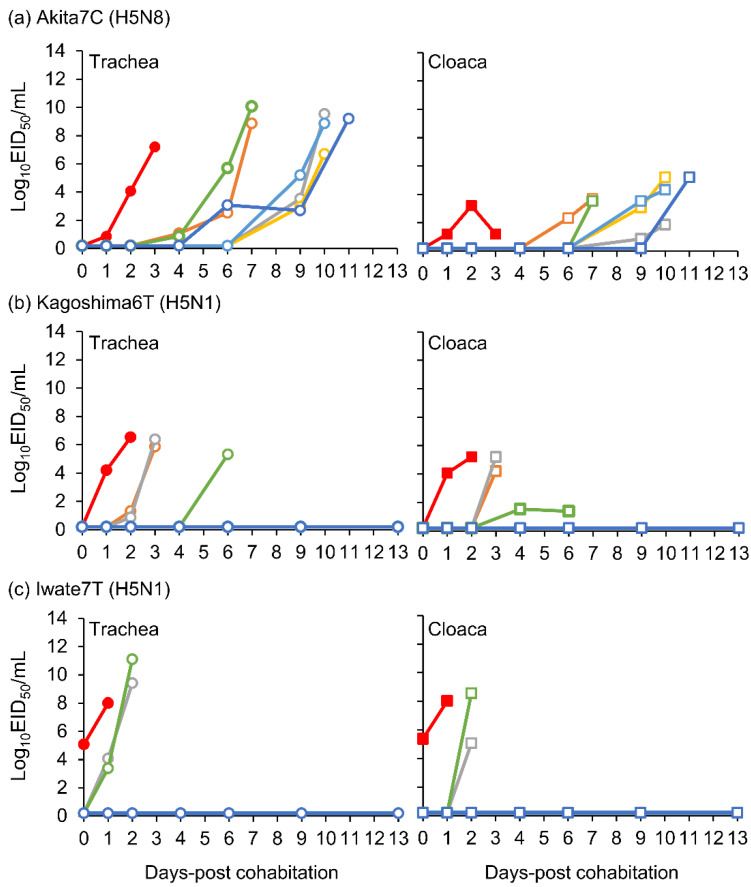
Viral titers in each tracheal and cloacal swab collected from a virus-inoculated chicken and cohabitated chickens ((**a**) Akita7C, (**b**) Kagoshima6T, and (**c**) Iwate7T strains). Viral titers in tracheal and cloacal swabs collected from virus-inoculated chickens are represented by red circles and squares, respectively. Viral titers in tracheal and cloacal swabs collected from cohabitated chickens are represented by blank circles and squares (green, orange, blue, light blue, gray, yellow), respectively.

**Table 1 viruses-15-00265-t001:** Chicken lethal dose, mean death time, mean maximum viral titer, and clinical signs.

Virus	CLD_50_ (log_10_EID_50_)	Chickens Inoculated with 10^6^ EID_50_ of Each Virus			Clinical Signs
		MDT	Mean Maximum Viral Titer (log_10_ EID_50_/mL)		
			Trachea	Cloaca	
Akita7C (H5N8)	3.83	3.5 days (82.8 h)	6.05 ± 1.41	2.99 ± 0.25	Depression (9/13), cyanosis in the comb, and subcutaneous hemorrhage in the legs (12/13)
Kagoshima6T (H5N1)	4.50	3.3 days (79.2 h)	5.97 ± 0.77	3.47 ± 0.40	Depression (9/10), neurological symptoms (2/10)
Iwate7T (H5N1)	4.68	2.2 days (52.8 h)	8.75 ± 0.99	7.51 ± 1.84	Depression (3/9), neurological symptoms (1/9)

**Table 2 viruses-15-00265-t002:** Transmission rate, mean death time, mean maximum viral titer, and clinical signs.

Virus	Transmission Rate (%)	Chickens Cohoused with the Inoculated Chicken			Clinical Signs	
		MDT	Mean Maximum Viral Titer (log_10_ EID_50_/mL)		Inoculated Chicken	Cohoused Chickens
			Trachea	Cloaca		
Akita7C (H5N8)	100 (6/6)	9.3 days (224 h)	8.87 ± 1.06	3.97 ± 1.14	Depression, cyanosis in the comb, and subcutaneous hemorrhage in the legs	Depression (3/6), cyanosis in the comb, and subcutaneous hemorrhage in the leg (6/6)
Kagoshima6T (H5N1)	50 (3/6)	4.2 days (100 h)	5.86 ± 0.43	3.64 ± 1.55	Depression	Depression (3/3), neurological symptoms (1/3)
Iwate7T (H5N1)	33.3 (2/6)	3.0 days (72 h)	10.25 ± 0.85	6.81 ± 1.74	Depression	Not observed (suddenly died)

**Table 3 viruses-15-00265-t003:** Serological cross-reactivity among H5 viruses by hemagglutination inhibition assay.

	Hyper Immune Antisera			
	ChibaC1T	Akita7C	Kagoshima6T	Iwate7T
ChibaC1T (H5N8)	160	640	20	320
Akita7C (H5N8)	80	320	20	160
Kagoshima6T (H5N1)	10	20	320	160
Iwate7T (H5N1)	80	320	40	320

## Data Availability

The data that support this study are available from the corresponding author upon reasonable request.

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
