# Peer review of "Different Infectivity and Transmissibility of H5N8 and H5N1 High Pathogenicity Avian Influenza Viruses Isolated from Chickens in Japan in the 2021/2022 Season"

_viruses, 2023, doi:10.3390/v15020265_

Round 1

Reviewer 1 Report

The manuscript entitled “Different infectivity and transmissibility of H5N8 and H5N1 high pathogenicity avian influenza viruses isolated from chickens in Japan in the 2021/22 season” by Takadate and colleagues aims at understanding the pathogenesis of 3 recent Japanese HPAIV H5Nx viruses. The study is very well written, clear, well-structured and presents important data for a better understanding of HPAI H5Nx pathogenesis in poultry. I just have the following questions and comments:

-          The authors used 2 different ages for birds used for IVPI. Both ages (4 and 7 weeks) are possible indeed for a valid IVPI assessment but do the authors have data suggesting identical results or may there be a difference?

-          Figure 2: why did the authors compare the 3 viruses with 10^6 EID50 as a dosis and not 1 or 10 CLD50? It would have made the comparison easier.

-          Lines 171-172: sentence to rephrase (CLD50 were not run with 10^6 EID50)

-          Tables 1 and 2: I feel 1 digit after the coma would be enough for the MDT (unless the authors really checked the embryos every 10 minutes?)

-          Figure 4: in the legend the authors could add that each color represents a contact bird to ease the reading.

-          Line 241: “crossly” should read “cross”.
Lines 292-293: it is unclear as such, please add ducks in the sentence.

Author Response

Reviewer #1

The manuscript entitled “Different infectivity and transmissibility of H5N8 and H5N1 high pathogenicity avian influenza viruses isolated from chickens in Japan in the 2021/22 season” by Takadate and colleagues aims at understanding the pathogenesis of 3 recent Japanese HPAIV H5Nx viruses. The study is very well written, clear, well-structured and presents important data for a better understanding of HPAI H5Nx pathogenesis in poultry. I just have the following questions and comments:

The authors used 2 different ages for birds used for IVPI. Both ages (4 and 7 weeks) are possible indeed for a valid IVPI assessment but do the authors have data suggesting identical results or may there be a difference?

Thank you for your comment. We don’t have any data to suggest whether results in this study are identical or not. Intravenous inoculation tests were urgently required to diagnose whether three influenza stains have high pathogenicity in chickens or not. Therefore, we could not use the same ages of chickens at that time. However, the same test has been conducted on index HPAIVs isolated in Japan using 4- to 8-week-old chickens since 2004, and the same results, lethality was 100%, were seen in all tests, suggesting results in this study are also identical.
The followings are articles describing intravenous inoculation tests;

Virology. 2005 Feb 5;332(1):167-76. doi: 10.1016/j.virol.2004.11.016.

Arch Virol. 2015 Jul;160(7):1629-43. doi: 10.1007/s00705-015-2428-9.

Transbound Emerg Dis. 2019 May;66(3):1227-1251. doi: 10.1111/tbed.13141.

Viruses. 2021 Mar 16;13(3):489. doi: 10.3390/v13030489.

Figure 2: why did the authors compare the 3 viruses with 10^6 EID50 as a dosis and not 1 or 10 CLD50? It would have made the comparison easier.

Thank you for your comment. In this study, we attempt to see the characterizes of three strains such as CLD50. The reason why we chose to show viral titers in each sample from chickens inoculated with 106 EID50 viruses is 106 EID50 viruses killed all chickens. We consider that comparing viral titer in the 106 EID50 ‘s group would make understanding easier.

Lines 171-172: sentence to rephrase (CLD50 were not run with 10^6 EID50)

Thank you for your suggestion. The sentence at lines 171-172 was changed to “CLD50 of three strains and MDT values of chickens inoculated with each of the three strains at the dose of 106 EID50 were calculated and summarized in Table 1.” in revised manuscript at lines 172-173.

Tables 1 and 2: I feel 1 digit after the coma would be enough for the MDT (unless the authors really checked the embryos every 10 minutes?)

Thank you for your comment. MDT values shown in Tables 1 and 2 shown at lines 178 and 219 were changed.

Figure 4: in the legend the authors could add that each color represents a contact bird to ease the reading.

Thank you for your comment. We changed the figure and the legend to make understanding easier.

Figure 4 is shown in revised manuscript at line 232.

Figure legend (at lines 233-238)

Viral titers in each tracheal and cloacal swab collected from a virus-inoculated chicken and cohabitated chickens ((a) Akita7C, (b) Kagoshima6T, and (c) Iwate7T strains). Viral titers in tracheal and cloacal swab collected from virus-inoculated chickens are represented by red circles and squares, respectively. Viral titers in tracheal and cloacal swab collected from cohabitated chickens are represented by blank circles and squares (green, orange, blue, light blue, gray, yellow), respectively.

Line 241: “crossly” should read “cross”.

Thank you for your comment. The word “crossly” at line 245 was changed with “cross” in the revised manuscript at line 245.

Lines 292-293: it is unclear as such, please add ducks in the sentence.

Thank you for your comment. The sentence at lines 292-293 was changed to “Three strains’ infectivity and transmissibility in ducks should be determined in future studies.” in the revised manuscript at lines 296-297.

Reviewer 2 Report

The following recommendations must be considered: 

1. An infographic or flow chart of the study must be part of the manuscript. 

2.  Refer to heading: 3.3. Clinical signs caused by three HPAI strains: Clinical signs and parameters must be presented in table. 

3.  Same comments stands for "3.4. Transmissibility of three HPAI strains in chickens./ Clinical signs of inoculated and cohoused chickens 204 were monitored daily for 14 days. "

4. Please support your statement by statistical p-value "In this study, we demonstrated that Akita7C has greater levels of infectivity and 317 transmissibility in chickens than the other strains tested,"

Consider the following reference for the improvement: 

Liang, Yuan, et al. "Pathogenesis and infection dynamics of high pathogenicity avian influenza virus (HPAIV) H5N6 (clade 2.3. 4.4 b) in pheasants and onward transmission to chickens." Virology 577 (2022): 138-148.

El-Shesheny, Rabeh, et al. "Highly Pathogenic Avian Influenza A (H5N1) Virus Clade 2.3. 4.4 b in Wild Birds and Live Bird Markets, Egypt." Pathogens 12.1 (2023): 36

Yang, Jing, et al. "Genetic, biological and epidemiological study on a cluster of H9N2 avian influenza virus infections among chickens, a pet cat, and humans at a backyard farm in Guangxi, China." Emerging Microbes & Infections 12.1 (2023): 2143282.

Author Response

Reviewer #2

The following recommendations must be considered:

  1. An infographic or flow chart of the study must be part of the manuscript.

Thank you for your suggestion. A flow chart of this study was added as Supplementally figure 1 at lines 454-459.

Supplementally figure 1 is added in the revised manuscript at line 454.

Figure legend (at lines 455-459)

Time schedule of sample collection during animal experiments. In the intranasal inoculation test, at 1, 2, 3, 5, 7, 10, and 14 dpi or at death, tracheal and cloacal swabs were collected. In the transmission study, at 0, 1, 2, 4, 6, 9, and 13 dpc or at death, tracheal and cloacal swabs were collected. Time points for the sample collection were represented by black allows. The time point for the cohabitation in transmission study was shown by a red allow.

The sentence at lines 109-110 “Tracheal and cloacal swabs were collected at 1, 2, 3, 5, 7, 10, and 14 days post inoculation (dpi) and at 0, 1, 2, 4, 6, 9, and 13 days post cohabitation (dpc) or at death (Supple-mentally figure 1).” in the revised manuscript at lines 109-111.

  1. Refer to heading: 3.3. Clinical signs caused by three HPAI strains: Clinical signs and parameters must be presented in table.

Thank you for your suggestion.

The title of table 1 was changed to “Table 1. Chicken lethal dose, mean death time, mean maximum viral titer, and clinical signs.” in the revised manuscript at line 178.

Information about clinical signs and parameters was added in Table 1.

The sentence “Clinical signs of chickens infected with three HPAI strains were also compared.” was changed to “Clinical signs of chickens infected with three HPAI strains were also compared (Table 1).” in the revised manuscript at lines 192-193.

  1. Same comments stands for "3.4. Transmissibility of three HPAI strains in chickens./ Clinical signs of inoculated and cohoused chickens 204 were monitored daily for 14 days. "

Thank you for your suggestion.

The title of table 2 was changed to “Table 2. Transmission rate, mean death time, mean maximum viral titer, and clinical signs.” in the revised manuscript at line 219.

Information about clinical signs and parameters was added in Table 2.

The following sentence was also added in the revised manuscript at lines 216-218 to describe clinical symptoms observed in the transmission study.

“Clinical signs observed in the transmission study were similar with those in the intranasal inoculation test (Table 2).”

  1. Please support your statement by statistical p-value "In this study, we demonstrated that Akita7C has greater levels of infectivity and 317 transmissibility in chickens than the other strains tested,"

Thank you for your comment. The infectivity and transmissibility of the three strains were not compared statistically in this study, since CLD50 and transmission rates could not be compared statistically. Instead of the infectivity and transmissibility of three strains, the survivability of chickens infected with three HPAIVs was compared statistically.

Consider the following reference for the improvement:

Liang, Yuan, et al. "Pathogenesis and infection dynamics of high pathogenicity avian influenza virus (HPAIV) H5N6 (clade 2.3. 4.4 b) in pheasants and onward transmission to chickens." Virology 577 (2022): 138-148.

El-Shesheny, Rabeh, et al. "Highly Pathogenic Avian Influenza A (H5N1) Virus Clade 2.3. 4.4 b in Wild Birds and Live Bird Markets, Egypt." Pathogens 12.1 (2023): 36

Yang, Jing, et al. "Genetic, biological and epidemiological study on a cluster of H9N2 avian influenza virus infections among chickens, a pet cat, and humans at a backyard farm in Guangxi, China." Emerging Microbes & Infections 12.1 (2023): 2143282.

Thank you for your comment. The articles written by Liang et al, and El-Shesheny et al were added as ref [35] and [15] in the revised manuscript at line 326 and 59.

The sentence in Discussion was changed to “Since the characteristics of HPAI strains in the 2021/22 season were not correlated with the number of outbreaks observed, relationships between the number of outbreaks and other factors such as the prevalence of HPAIVs in wild birds including residential birds [35] should be addressed in future studies.” in the revised manuscript at lines 323-327.

Reviewer 3 Report

Three different types of HPAIV entered Japan during the 2021-2022 season. Takadate et al describes on the comparison of infectivity in chickens, and clinical signs and viral shedding in chickens after intranasal inoculation, using chicken isolates representing three types. In addition, the transmissibility between chickens was also evaluated for each virus. The results seem to contradict the author’s assumptions, but other possibilities were well discussed.

The present study was properly conducted and contains important information.

Although there is nothing in particular to point out, I would like to confirm one point: do 20A, 20E and 21E refer to genotypes or are they simply HA lineages? 

Author Response

Reviewer #3

Three different types of HPAIV entered Japan during the 2021-2022 season. Takadate et al describes on the comparison of infectivity in chickens, and clinical signs and viral shedding in chickens after intranasal inoculation, using chicken isolates representing three types. In addition, the transmissibility between chickens was also evaluated for each virus. The results seem to contradict the author’s assumptions, but other possibilities were well discussed. The present study was properly conducted and contains important information.

Although there is nothing in particular to point out, I would like to confirm one point: do 20A, 20E and 21E refer to genotypes or are they simply HA lineages?

Thank you for your comment. The 20A, 20E and 21E are groups based on HA lineages, not genotypes.
